



# Effects of vegetation and soil on evapotranspiration, flow regime, and basin storage in three nearby catchments in northeast Japan

Shoji Noguchi[1], Tomonori Kaneko[2], Shin'ichi Iida[1], Wataru Murakami[1], Takanori Shimizu[1]

[1] Forestry and Forest Products Research Institute, 1 Matsunosato, Tsukuba, Ibaraki, 305-8687, Japan

[2] Akita Forestry Research and Training Center, 47-2 Idoshiridai,Kawabe-Toshima, Akita 019-2611, Japan

*Correspondence to*: Shoji Noguchi (noguchi@ffpri.affrc.go.jp)

**Abstract**

Vegetation and soil determine evapotranspiration, flow regime, and basin storage in forested catchments. We conducted hydrological observations at three nearby catchments (catchments nos. 1, 2, and 3) in the Nagasaka experimental watershed

located on the green tuff region in northeast Japan. Diameter-at-breast height (DBH) of all trees >3 cm DBH was recorded. In addition, we measured soil depth at 170 locations and investigated 45 soil pits. Based on these detailed vegetation and soil measurements, we examined evapotranspiration, flow regime, and basin storage during the no-snow-cover period (May–November). More than 80.9% of stands in the catchment were comprised of *Cryptomeria japonica*. Stand volume (122.0 m$^3$ ha$^{-1}$) and sapwood area (10.7 m$^2$ ha$^{-1}$) in catchment no. 3 were smaller than those in the other two catchments (no. 1: 255.7

m$^3$ ha$^{-1}$; 16.0 m$^2$ ha$^{-1}$, no. 2: 216.5 m$^3$ ha$^{-1}$; 14.2 m$^2$ ha$^{-1}$). Consequently, evapotranspiration was lower in catchment no. 3 than that in catchments nos. 1 and no. 2. In addition, low and scanty runoffs in catchment no. 3 were larger than those in nos. 1 and 2. The order of magnitude for soil storage was catchments no. 1 (104.2 mm) < no. 3 (115.7 mm) < no. 2 (123.1 mm), which was similar to the order of magnitude for basin storage: catchments no. 1 (65.9 mm) < no. 3 (69.7 mm) < no. 2 (115.8 mm). Deep soil storage contributed to increased basin storage and decreased the ratio of plentiful runoff to scanty runoff.

**Keywords.** Catchment scale, *Cryptomeria japonica*, Nagasaka experimental watershed, Sapwood area, Soil storage, Stand volume

## 1 Introduction

Forests have multiple important functions, including biodiversity conservation, global environmental protection, headwater conservation and recharge improvements, and forest production, among others (Science Council of Japan 2001). Two-thirds

of Japan's land area is covered by forest, and approximately 40% of all forests in Japan are artificially planted. Promoting forest management and forest conservation is important to realize the full potential of forests (Forest Agency, 2016).

Forest-watershed experiments have been conducted to evaluate the function of head water conservation and recharge improvements (Nakano 1971; Tsukamoto 1998). Evapotranspiration decreases after vegetation cover decreases following forestry operations, such as clear cutting (Fujieda et al. 1996; Maita and Suzuki 2007), counter line strip cutting (Shimizu et



al. 1994), and thinning (Kubota et al. 2013). Declines in forested area (e.g. forest fires, pine wilt disease, or clear cutting) affect the flow regime (Tani and Abe 1987; Tamai et al. 2004; Maita and Suzuki 2008). These studies on flow regime reported that the increase ratio of discharge was higher at low flow than at high flow. Thus, vegetation is one of the most important factors determining evapotranspiration and flow regime. Interestingly, Onda (1992) reported that peak flow in a

catchment underlain by granodiorite bedrock is smaller than that in a catchment underlain by granite bedrock because soil depth of the former is deeper than that of the latter. Additionally, rain water is retained in soil and only a small proportion appears as stormflow under dry conditions; stormflow volume increases with increasing soil moisture (Noguchi et al. 1997; Tani et al. 2012). These results suggest that soil storage is another important factor determining runoff characteristics.

Evapotranspiration, flow regime, and basin storage are very difficult to determine under various climates because of

differences in geology, topography, soil, and vegetation. Paired catchment studies have evaluated changes in vegetation based on the magnitude of water yield changes (Bosch and Hewlett 1982; Brown et al. 2005). Regardless of the kinds of trees (e.g. conifer, hardwood, eucalyptus, or scrub), the change in water yield increases with increasing percentage forest cover. In addition, the regression between the change in water yield and percentage increase in forest cover is highly correlated in shrub but is poorly correlated in conifer, hardwood, eucalyptus forests. (Bosch and Hewlett 1982; Brown et al.

2005). This is because water yield changes as coniferous (Murakami et al. 2000) and eucalyptus forest (Cornish and Vertessy 2001; Vertessy et al. 2001) stands age. Thus, percentage changes in forest cover and vegetation status are needed more detail to evaluate water yield. Moreover, although detailed catchment-scale topographical data of the forest area can be generated by airborne laser scanner (Setojima et al. 2002; James et al. 2007), it remains extremely difficult to investigate soil at a catchment scale. In addition, insufficient vegetation and soil data are available compared to those based on topographical

information.

Vegetation and soil are important factors determining hydrological properties, such as evapotranspiration, flow regime, and basin storage in forested catchments. We conducted hydrological observations in three nearby catchments with similar topography. We also measured soil physical properties and investigated vegetation in the three catchments. Our objectives were: (1) to clarify vegetation and soil at a catchment scale; (2) to compare evapotranspiration, flow regime, and basin

storage among the three catchments; (3) to examine the relationships between hydrological factors (: vegetation and soil) and hydrological characteristics (: evapotranspiration, flow regime, and basin storage).

## 2 Material and methods

### 2.1 Site description

Observations and investigation were made at the Nagasaka experimental watershed (NEW; 40°16′N, 140°24′E, Akita

Prefecture in northern Japan; Fig. 1). The NEW has three catchments: no.1 (6.55 ha, elevation 110–168 m), no.2 (7.52 ha; elevation, 88–174 m), and no.3 (6.50 ha; elevation, 84–152 m). Surficial geology is Neogene tuffaceous rock (green tuff) that has been weathered and softened (Kaneko et al., 2010). The two principal soil series are low-humic andosol and



moderately moist brown forest soil. Mean slope gradients in catchment nos.1, 2, and 3 were 19.5°, 22.0° and 22.0°, respectively. The ratio of all slopes in no.1 catchment was 12.3%–17.6% except for the southwest- (6.8 %) and west- (2.5%) facing slopes. While the east- (19.7%) and southeast- (23.5%) facing slopes had higher ratios in catchment no. 2, the ratios of the south-, southwest- and west- facing slopes were ≤10%. The southeast- (19.9%) and northwest- (20.1%) facing slopes

in catchment no. 3 had higher ratios. However, the ratios for the southwest- and west-facing slopes were < 5% (Fig. 2).
The catchments were mainly covered with *Cryptomeria japonica* that had been planted in 1963 in catchment nos. 1 and 2 and in 1970 in catchment no. 3. Salvage cutting was conducted in all three catchments between 1987 and 1995. The catchment no. 1 hillsides supported mixed deciduous broad-leaved forest with *Querus serrata* and *Quercus crispula*, etc. Some ridge lines were inhabited by *Pinus densiflora*. In addition, part of catchment no. 1 was covered with *Chamaecyparis*

*obtusa* stands. Various understory species, such as *Arachniodes standishii*, *Carex curvicollis*, and *Rhododendron kaempferi* var. *kaempferi*, coexisted in complex patterns within the coniferous forests.
Annual mean precipitation was 1,905.7 mm, during 2007–2009. Daily mean air temperature was between −7.2°C –29.4°C (mean, 10.3°C) (Noguchi et al. 2007). The nearest Automatic Meteorological Data Acquisition System data site at Takanosu (40°13.6′N, 140°22.2′E, elevation: 29 m), which was 7 km southwest of the NEW, where mean annual air temperature,

rainfall, and maximum snow depth during 1981–2010 were 10.2°C, 1671.1 mm. and 75.2 cm, respectively (data from the Japan Meteorological Agency website). We compared rainfall, mean air temperature, and sunshine duration during May–November 2004–2006 and mean (1981–2010) data at the AMeDAS site. Rainfall ranged from 958 to 1,277 mm and mean rainfall (1,134 mm) was similar to the 1981–2010 mean (1,120 mm). Mean air temperature for the period was 16.9°C–17.1°C and was higher every study year than mean (16.3°C). Conversely, cumulative sunshine duration was 783.6–1,027.5

hours and was lower during all years than mean (1,081.1 hours).

## 2.2 Tree census and sapwood area

Diameter-at-breast height (DBH) of all trees ≥3 cm was recorded, and these trees were divided into conifers (: *C. japonica*, *Pinus densiflora*, and *C. obtusa*) and deciduous broad-leaved in the three catchments in 2001. We selected representative trees to measure height using a laser distance meter (Impulse LT2000; Laser Technology Inc., Centennial, CO, USA) in each

catchment. An approximate expression for the relationship between tree height and DBH was obtained, based on the measurements. Tree height of was estimated using these equations when measurements were not available. We determined the width of the sapwood by the color difference between the sapwood and heartwood cased on core sampling of wood by increment borer in 16 *C. japonica* stands.
The amount of transpiration from a stand was calculated using mean sap flux density and total sapwood area (e.g. Kumagai

et al., 2007; Iida et al., 2015). We estimated transpiration by evergreen conifers based on sapwood area (*AS*). Tsuruta et al. (2011) determined the allometric relationship between *AS* and DBH in *C. japonica* stands located in the districts of Kyusyu, Hokuriku and Kanto, Japan. In addition, they argued that Eq. (1) could be used to determine a common allometric relationship at the same level of precision as with the site-specific equation.





$$AS = 10.0DBH - 41.5(DBH \leq 30\ cm)\ and\ AS = 19.1DBH - 314.8\ (DBH > 30\ cm) \tag{1}$$

We obtained the following Eq. (2) for the relationship between $AS$ and DBH in *C. japonica* stands in the NEW:

$$AS = 12.6DBH - 94.9, R^2 = 0.909 \tag{2}$$

Sapwood area increased with increasing DBH (Fig. 3). The root mean square errors between the measured and estimated

values using Eqs. (1) and (2) were 35.2 and 21.9, respectively. Then, we calculated $AS$ in *C. japonica* stands using Eq. (2). In addition, we calculated $AS$ of *C. obtusa* stands using Eq. (3) in Tsuruta et al. (2011) and $AS$ of *Pinus densiflora* stands using Eq. (4) in Iida et al. (2006):

$$AS = 8.2DBH - 27.4(DBH \leq 32\ cm)\ and\ AS = 19.1DBH - 374.7\ (DBH > 32cm) \tag{3}$$

$$AS = 3.52DBH^{1.302} \tag{4}$$

**2.3 Measurement of soil physical properties**

Soil depth was measured at 10m intervals along five or six lines from the main stream to the ridge (Fig. 2) using a Hasegawa soil sampler (Daitou Techno Green, Inc., Tokyo, Japan) until the 1m mark was reached. Fifty-seven, 56, and 57 measurement points were taken at catchments nos. 1, 2, and 3, respectively (total, 170 points). Fifteen soil pits were selected along the soil depth measurement lines in each catchment (totals, 45 pits) (Fig. 2). Undisturbed soil cores were collected at

layers A, B and C in each soil pit. The saturated hydraulic conductivity of the cores was measured using a constant head permeameter (Mashimo 1960). The percentage maximum water holding capacity (WHC) was measured using the weight after the soil cores were saturated. Fine size pores (FSP; pF < 2.7) in the soil (%) were measured using a porous plate (Mashimo 1961). Coarse size pores (CSP) in the soil (%) and soil water storage (SWS) (mm) were calculated (Fujieda 2007) as follows:

$$CSP = WHC - FSP \tag{5}$$

$$SWS = \frac{CSP}{100} \times 10Sd \tag{6}$$

Where $Sd$ is soil depth (cm).

**2.4 Hydrological observations and data**

Precipitation was measured using a tipping-bucket rain gauge (B-071, Yokogawa Denshikiki Co., Ltd., Tokyo, Japan) at a height of 3m at the meteorological station (elevation, 100 m; Fig. 1). The discharge amount from all three catchments was observed using a single 60° V-notch flow-gauging weir with a $3 \times 4$m sand basin. Water levels were observed using the same pressure-type water-level gauges (KDC-S10-D, Kona System, Sapporo, Japan). Discharge and precipitation data for 2004–2006 were used during the no-snow-cover period (May–November) in this study.


**2.5 Hydrological analysis**

**2.5.1 Evapotranspiration**




Evapotranspiration was estimated using the short-time period water-budget method. The relationship between water storage in basin $S(t)$ and discharge rate $q(t)$ can be written using Eq. (7):

$$S(t) = f[q(t), dq/dt] \tag{7}$$

Assuming the water storage values $S(t_1)$ and $S(t_2)$ are equal when $q(t)$ and $dq/dt$ become equal at times $t_1$ and $t_2$, the change in

water storage $\Delta S = S(t_2) - S(t_1)$ is zero. $ET$ was calculated using Eq. (8):

$$ET = P - Q = \int_{t_2}^{t_1} p(t)dt - \int_{t_2}^{t_1} q(t)dt \tag{8}$$

Where $P$ and $Q$ are total precipitation and total discharge from $t_1$ to $t_2$, and $p(t)$ and $q(t)$ are the rainfall intensity and discharge rate, respectively. The following procedure proposed by Suzuki (1985) and Noguchi et al. (2004) was used to determine the water budget periods for $t_1$ and $t_2$.

1)    Dates with ≤2.0 mm d$^{-1}$ of precipitation over 2 days or ≤2.0 mm d$^{-1}$ of precipitation were chosen as dates for $t_1$ or $t_2$, respectively.
2)    Based on these dates, pairs of dates between which the difference in the daily discharge rate was ≤5 % were selected.
3)    Pairs of dates for which the intervening period was ≤10 days or ≥60 days were excluded.

**2.5.2 Flow-duration curve and flow regime**

A flow-duration curve was plotted for the experimental period (214 days from May to November during the no-snow season) using daily runoff data. The curve was divided into four sections. The most runoff over 82 days was designated as plentiful runoff. Total runoff values for the subsequent 52, 50, and remaining 30 days were designated as ordinary, low, and scanty runoff, respectively. The numbers of days for the various runoff types were distributed using the proportions of the various

runoff types in the annual flow-duration curve (Nakano et al. 1963). The ratio of plentiful to scanty runoff was used as an index to represent the function of headwater recharge (Shimizu 1980).

**2.5.3 Basin storage**

Hourly discharge was plotted continuously on a semilogarithmic scale. A point of inflection was obtained on the falling limb

of the hydrograph between 12 and 72 h after a storm. Stormflow was defined as the area above the separation line; that is, the line on the hydrograph that connects the point of rise to the point of inflection.

Water loss, $L$ (mm), was computed based on stormflow, $SF$ (mm), and precipitation, $P$ (mm), for every storm event using Eq. (9):

$$L = P - SF \tag{9}$$

$L$ increases with increase in $P$ and can be similar with the upper limit when the curve converged. The curve was defined as a retention curve using Eq. (10):

$$L = S_B(1 - e^{-kP}) \tag{10}$$

Where $S_B$ is basin storage (mm) and $k$ is a constant (Endo 1985; Fujieda 2007).

**2.5.4 Statistics**





Statistical analyses were performed to test for differences in soil depth variations in the three catchments using Scheffe's post-hoc test. A P-value <0.001 was considered significant. These analyses were conducted using SPSS ver. 20.0 software (SPSS Inc., Tokyo, Japan).

## 3 Results

### 3.1 Vegetation

The ratios of evergreen coniferous tree stand volume (ratio of *C. japonica* stand volume) in catchments nos. 1, 2, and 3 were 96.0% (80.9%), 99.6% (97.5%), and 94.7% (92.8%), respectively. *P. densiflora* was distributed on the ridge of each catchment. *C. obtusa* occurred only in catchment no. 1. *Q. serrata*, *Q. crispula* Blume, and *Acanthopanax sciadophylloides* were fairly well represented as deciduous broad-leaved trees. In particular, catchment no. 3 supported many small diameter deciduous broad-leaved trees. *C. japonica* was the principal tree in each catchment. The frequency of occurrence of the *C. japonica* stands is shown in Fig. 4a. Although stand density was 1.6 times larger in catchment no. 3 (2,050 stand ha$^{-1}$) than that in catchment nos. 1 (1,260 stand ha$^{-1}$) and 2 (1,249 stand ha$^{-1}$), *C. japonica* stand volume in catchment no. 3 (113.2 m$^3$ ha$^{-1}$) was nearly half that of catchment nos. 1 (206.9 m$^3$ ha$^{-1}$) and 2 (211.1 m$^3$ ha$^{-1}$). Total stand volumes in catchment nos. 1, 2, and 3 were 255.7, 216.5, and 122.0 m$^3$ ha$^{-1}$, respectively (Fig. 4b).

Figure 5 shows total sapwood area in each catchment. The order of magnitude for total *C. japonica* sapwood area was: catchment no. 3 (10.6 m$^2$ ha$^{-1}$) < no. 1 (13.1 m$^2$ ha$^{-1}$) < no. 2 (13.8 m$^2$ ha$^{-1}$). The order of magnitude for the total evergreen coniferous tree sapwood area was: catchment no. 3 (10.7 m$^2$ ha$^{-1}$) < no. 2 (14.2 m$^2$ ha$^{-1}$) < no. 1 (16.0 m$^2$ ha$^{-1}$); total sapwood area of catchment no. 3 was significantly smaller than that of the other two catchments. The amount of transpiration from a catchment is the product of mean sap flux density and total sapwood area.

### 3.2 Soil physical properties

Table 1 summarizes the soil physical properties in each catchment. The very coarse size pore and saturated hydraulic conductivity decreased with increasing soil depth. Mean soil depth (± standard error) in catchment nos. 1, 2, and 3 were 65.0 (±2.5), 79.2 (±3.0), and 74.2 (±2.6) cm, respectively. The differences in mean soil depth between catchment nos. 1 and 2, and between catchment nos. 3 and 2 were significant. Furthermore, the numbers of sampling points with ≥100 cm soil depth (measurement limit of the Hasegawa soil sampler) were 2, 15 and 5 points in catchment nos. 1, 2, and 3, respectively. Soil water storage in soil layer A + B in catchment nos. 1, 2, and 3 were 104.2, 123.1, and 115.7 mm, respectively.

### 3.3 Evapotranspiration, flow regime, and basin storage

Figure 6 indicates estimated monthly evapotranspiration using the short-time period water-budget method. Evapotranspiration peaked in August in each catchment. Evapotranspiration in catchment no. 3 tended to be lower than that





in the other two catchments. The evapotranspiration during June–October was in the order of magnitude: catchment no. 3 (463.2 mm) < no. 1 (510.2 mm) < no. 2 (538.9 mm).

Table 2 shows the flow regime for the three experimental periods (May–November 2004–2006) in the three catchments. Low and scanty runoff in the NEW catchment increased as the amount of rainfall increased during the 3 years in the following order: 2004 > 2005 > 2006 (Table 2). The flow-duration curves were higher as the amount of rainfall increased within all catchments during the study period. Daily discharge by catchment no. 3 was larger than that by catchment nos. 2 and 1. The ratio of plentiful to scanty runoff in catchment no. 1 (52.6) was relatively higher than the ratios for catchment nos. 2 (33.6) and 3 (36.2).

Figure 7 shows the relationships between storm period rainfall and water loss in the three catchments. When the initial runoff was ≥0.1 mm h$^{-1}$, the basin storage order of magnitude in each catchment was: no. 1 (28.0 mm) < no. 3 (28.5 mm) < no. 2 (56.7 mm). When initial runoff was ≤0.02 mm h$^{-1}$, the basin storage order of magnitude in each catchment was: no. 1 (84.1 mm) < no. 3 (88.5 mm) < no. 2 (162.2 mm). When initial runoff was smaller, the basin storage became larger. Based on all data, the order basin storage of magnitude in each catchment was: no. 1 (65.9 mm) < no. 3 (69.7 mm) < no. 2 (115.8 mm).

## 4 Discussion

Evidence that geology have an effect on the flow regime in mountainous areas can be seen, namely runoff was relatively flashy in sedimentary rock basins but stable in igneous rock basins (Shimizu 1980; Tani et al. 2012). Surficial geology at the three catchments in this study is the green tuff, which is distributed over all six prefectures of northeast Japan, and occupies most of Yamagata and Akita Prefectures (Kitamura, 1983). Therefore, the flow regime can be discussed in relation to the geological features in northeastern, Japan. The gradient and aspect of slopes were similar among the three catchments (Fig. 2). Consequently, differences in the effects of topography on evaporation, flow regime, and basin storage were small among the three catchments. Thus, the effects of vegetation and soil on evapotranspiration, flow regime, and basin storage were considered in three catchments with the same geology and similar topography.

### 4.1 Effects of vegetation on evapotranspiration and flow regime

Evapotranspiration generally decreases as forested area decreases (Bosch and Hewlett 1982; Brown et al. 2005). Kubota et al. (2013) reported 21.3% (113.8 mm over 5 months) reduction in evapotranspiration after thinning (49.5% of stems removed) *C. japonica* and *C. obtusa* forests. Strip cutting (27.4% of stems and 54.0% of stand volume removed) caused a 9.6% (63.1 mm over 5 months) decrease in evapotranspiration in a mixed forest (e.g. *Thujopsis dolabrata*, *Fagus* spp, and *Quercus* spp: Shimizu et al. 1994). Although *C. japonica* stand density in our study was 1.6 times larger in catchment no. 3 than that in other two catchments (Fig. 4b), stand volume in catchment no. 3 was almost half as much as that in catchment nos. 1 and 2 (Fig. 4a). On the other hand, evapotranspiration was lower in catchment no. 3 than that in nos. 1 (9.2% reduction; 47.0 mm



for 5 months) and no. 2 (14.1% reduction; 75.8 mm for 5 months) (Fig. 6). These results suggest that stand volume has more of an effect on evapotranspiration than stand density.

Canopy interception and transpiration are the main factors affecting evapotranspiration in a forest. The water balance method as well as other methods (e.g. interception, sap flow, flux measurements) is effective for evaluating evapotranspiration

quantitatively (Kosugi et al. 2007; Shimizu et al. 2015). The mean ratios of throughfall to rainfall over 570 days in each NEW catchment were 82.1% (no.1), 80.1% (no.2), and 80.3% (no.3), respectively (Iwaya et al. 2013). In general, the ratio of stemflow to rainfall was smaller than the ratio of throughfall to rainfall. These results suggest that the difference in interception loss among the three catchments was relatively small. Currently, we do not have any data related to sap flux density in the NEW. Kajitani et al. (2005) pointed out that diurnal changes in *C. japonica* sap flow differ between the upper

and lower parts of a slope because of differences in micrometeorological conditions among various locations. We found no differences related to the direction or degree of slope among the three catchments (Fig. 2), suggesting that little difference exists in the micrometeorological conditions among the three catchments. Thus, we assumed that catchment-to-catchment differences in mean sap flux density had a minimal effect on the difference in transpiration among catchments. On the other hand, total sapwood area in catchment no. 3 was smaller than that in catchment nos. 1 and 2 (Fig. 5). These results suggest

that the transpiration in catchment no. 3 could be smaller than that in catchment nos. 1 and 2. Kumagai et al. (2014) also reported the same results in a 50-year-old *C. japonica* forest: although tree density in the upper slope plot (UP) was higher than that in the lower slope plot (LP), sapwood area and transpiration in the UP was lower than that in the LP. These results agree with our results of monthly evapotranspiration within each catchment (Fig. 6).

Flow regime depends on the geological features of a catchment and can be calculated using discharge data from a multiple

purpose dam and a gauging station along the river (Shimizu 1980). However, the precise relationship between vegetation in a catchment and flow regime could not be obtained by examining these large catchments, which have several dozens to several hundred km$^2$ of cover (Shimizu 1980). In contrast, decreasing the forested area decreases evapotranspiration and increases in low and scanty runoff in small experimental catchments (Tani and Abe 1987; Tamai et al. 2004; Maita and Suzuki 2008). Low and scanty runoff were also lower due to the higher evapotranspiration rate in the present study (Table 2; Fig. 6).

Evapotranspiration and flow regime depend on forest stand age (Murakami et al. 2000; Cornish and Vertessy 2001; Vertessy 2001; Vertessy et al. 2001). Several studies have investigated vegetation using airborne LiDAR data (Chasmer et al. 2006; Wasser et al. 2013) and aerial images from an unmanned aerial vehicle (UAV) (Gini et al. 2012; Sakai et al. 2016) in a catchment. In particular, relatively simple and inexpensive vegetation information can be obtained using UAV. Therefore, we propose recurring vegetation investigations using an UAV regularly to evaluate the effect of vegetation on

evapotranspiration and flow regime in the NEW.

### 4.2 Effects of soil on basin storage and flow regime

Initial runoff related to stormflow analysis is an index that describes soil moisture conditions of a catchment; many studies have reported that the relationship between the amount of rainfall and stormflow depends on the initial runoff (e.g. Tani and



Abe 1987; Noguchi et al. 2005; Siti et al. 2014). Basin storage in the present study also depended on initial runoff (Fig. 7). The differences in basin storage between initial runoff $\leq 0.02$ mm h$^{-1}$ and initial runoff $\geq 0.1$ mm h$^{-1}$ in each catchment were 56.1 mm (no. 1), 105.5 mm (no. 2), and 60.0 mm (no. 3). Soil moisture conditions of the catchment (e.g. initial runoff) must be considered when basin storage is compared among several catchments. Although basin storage in each catchment

depended on the initial runoff, the order of magnitude for basin storage for the three catchments was the same. The order of magnitude for basin storage in each catchment (Fig. 7) was the same as the order of magnitude for soil storage (Table 1). These results suggest that higher soil storage helps decrease stormflow.

The ratio of plentiful to scanty runoff is used as an index to represent the function of headwater recharge, and a low value suggests a strong effect of headwater recharge in a catchment (Shimizu 1980). This ratio was larger in catchment no. 1 than

those in catchment nos. 2 and 3 (Table 2) and soil storage in catchment no. 1 was relatively smaller than those in nos. 2 and 3 (Table 1). In contrast, the plentiful to scanty runoff ratio in catchment no. 2 was low, and soil storage was high. These results suggest that headwater recharge has more of an effect in a catchment with greater soil storage. We showed that lower scanty runoff (Table 2) was caused by high evapotranspiration. In that case, soil storage and vegetation have an effect, which is why the ratio in catchment no. 1 was relatively larger.

The Hasegawa soil sampler can only measure a soil depth of 100 cm. Therefore, we evaluated soil storage only in the A + B soil layers in this study. However, 2, 15, and 5 of the sampling sites were >100 cm deep in catchment nos. 1, 2, and 3, respectively. In addition, the C layer was very porous (Table 2), suggesting that it is important to evaluate soil storage at >100 cm deep in a future study.

**5 Author contributions**

Noguchi, S. performed the research, analysed the data, and wrote the manuscript. Kaneko, T. conceived the study, performed the research, and contributed to the soil properties analyses and writing. Iida, S and Shimizu, T. performed the research and contributed to writing. Murakami, W. contributed to GIS analyses and the writing.

**6 Competing interests**

The authors declare no conflicts of interest.

**7 Acknowledgements and Funding Information**

The authors thank all personnel at the Akita Forestry Research and Training Center for their assistance with data collection.

This study was financially supported by the Japan Society for the Promotion of Science (JSPS) for KAKENHI (Grant No. 23221009) and Forestry and Forest Products Research Institute.



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





**Table 1 Soil physical properties**

| Catchment | Soil layer | *Thickness of soil layer (cm) | Soil water storage (mm) | *Soil layer | *Maximum water holding capacity (%) | *Coarse size pores (%) | *Fine size pores (%) | **Saturated hydraulic conductivity (cm s$^{-1}$) |
|---|---|---|---|---|---|---|---|---|
| No.1 | A | 25.9 | 35.7 | A | 62.0 | 13.8 | 48.2 | $1.18\times10^{-2}$ |
| | n = 57 | (1.3) | | n = 15 | (2.2) | (0.7) | (2.2) | |
| | ***B | 39.1 | 68.4 | B | 60.6 | 17.5 | 43.1 | $1.46\times10^{-3}$ |
| | n = 57 | (2.3) | | n = 15 | (1.7) | (1.4) | (1.2) | |
| | ***A+B | 65.0 | 104.2 | C | 60.4 | 12.4 | 48.0 | $2.23\times10^{-4}$ |
| | n = 57 | (2.5) | | n = 15 | (1.9) | (1.5) | (1.7) | |
| No.2 | A | 35.4 | 53.5 | A | 59.0 | 15.1 | 44.0 | $1.15\times10^{-2}$ |
| | n = 56 | (1.8) | | n = 15 | (1.9) | (0.7) | (2.1) | |
| | ***B | 43.8 | 69.6 | B | 58.3 | 15.9 | 42.4 | $1.17\times10^{-3}$ |
| | n = 56 | (3.1) | | n = 15 | (1.7) | (1.7) | (1.1) | |
| | ***A+B | 79.2 | 123.1 | C | 52.1 | 7.9 | 42.9 | $4.38\times10^{-4}$ |
| | n = 56 | (3.0) | | n = 15 | (2.3) | (1.0) | (1.7) | |
| No.3 | A | 30.1 | 50.9 | A | 61.8 | 16.9 | 44.9 | $1.79\times10^{-2}$ |
| | n = 56 | (2.3) | | n = 15 | (1.3) | (1.4) | (2.1) | |
| | ***B | 44.1 | 64.8 | B | 55.6 | 14.7 | 40.9 | $2.01\times10^{-3}$ |
| | n = 56 | (2.6) | | n = 15 | (1.9) | (1.1) | (1.8) | |
| | ***A+B | 74.2 | 115.7 | C | 51.6 | 9.2 | 42.5 | $8.07\times10^{-4}$ |
| | n = 56 | (2.6) | | n = 15 | (2.5) | (0.8) | (2.6) | |

\* Arithmetic mean

\*\* Geometric mean

\*\*\*Two, 15, and five sampling points at depths >100 cm soil deep in catchment no. 1,2, and 3, respectively.

5    n, number of samples

Numbers in parenthesis are standard errors.





**Table 2 Plentiful, ordinary, low, and scanty runoff in catchment nos. 1, 2, and 3 of the Nagasaka experimental watershed.**

| Period | Rainfall (mm) | Catchment | Plentiful runoff (mm) | Ordinary runoff (mm) | Low runoff (mm) | Scanty runoff (mm) | Ratio of plentiful to scanty runoff |
|---|---|---|---|---|---|---|---|
| May– Nov. in 2004 | 1487.0 | No.1 | 707.8 | 102.5 | 57.8 | 18.0 | 39.3 |
| | | No.2 | 588.9 | 110.7 | 62.6 | 24.2 | 24.3 |
| | | No.3 | 726.8 | 120.4 | 66.9 | 23.6 | 30.8 |
| May– Nov. in 2005 | 1344.0 | No.1 | 526.0 | 77.4 | 33.3 | 6.9 | 76.2 |
| | | No.2 | 467.9 | 88.0 | 37.7 | 8.5 | 55.0 |
| | | No.3 | 585.5 | 95.5 | 48.8 | 13.1 | 44.7 |
| May– Nov. in 2006 | 1116.0 | No.1 | 374.2 | 64.2 | 23.6 | 5.8 | 64.5 |
| | | No.2 | 327.3 | 56.2 | 26.4 | 8.6 | 38.1 |
| | | No.3 | 403.3 | 73.4 | 34.5 | 10.7 | 37.7 |
| Mean | 1315.7 | No.1 | 536.0 | 81.3 | 38.3 | 10.2 | 52.5 |
| | | No.2 | 461.4 | 85.0 | 42.2 | 13.7 | 33.7 |
| | | No.3 | 571.9 | 96.4 | 50.1 | 15.8 | 36.2 |

Largest total runoff over 82 days was designated plentiful runoff. Total runoff for the remaining 52, 50, and 30 days were designated

5  ordinary, low, and scanty runoff respectively.





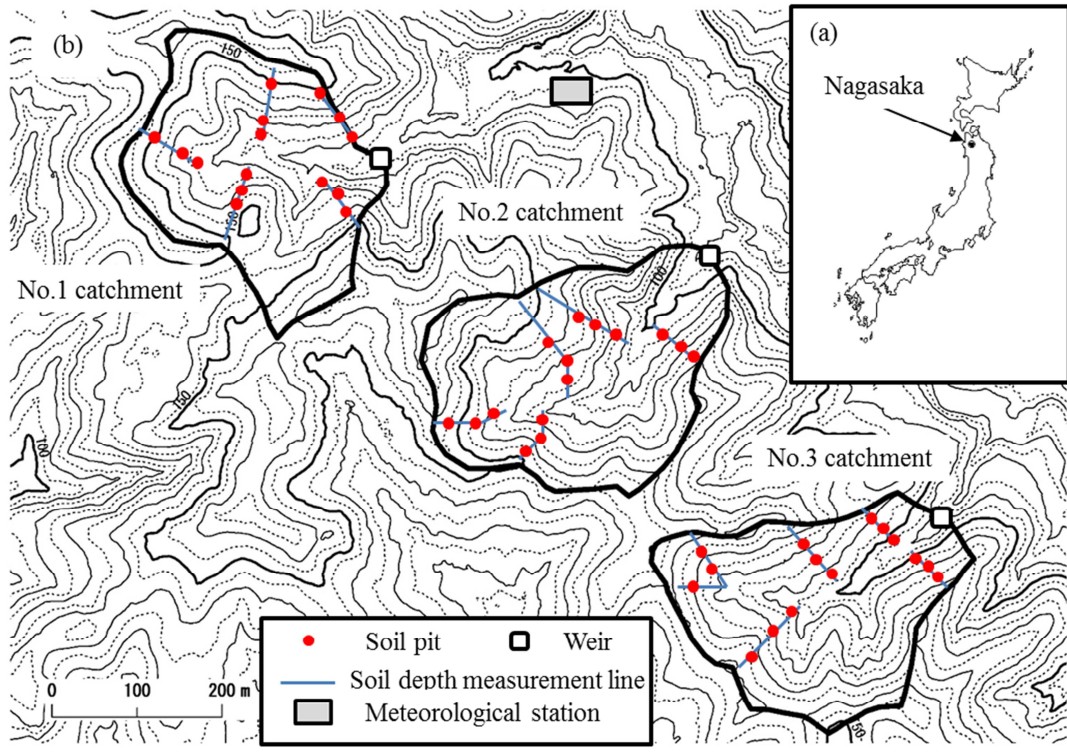

5    **Figure 1: Map of the study site. (a) Location of the Nagasaka Experimental Watershed (NEW) in Akita Prefecture, Japan. (b) Topography and locations of the soil survey lines and soil pits at three catchments in the NEW.**





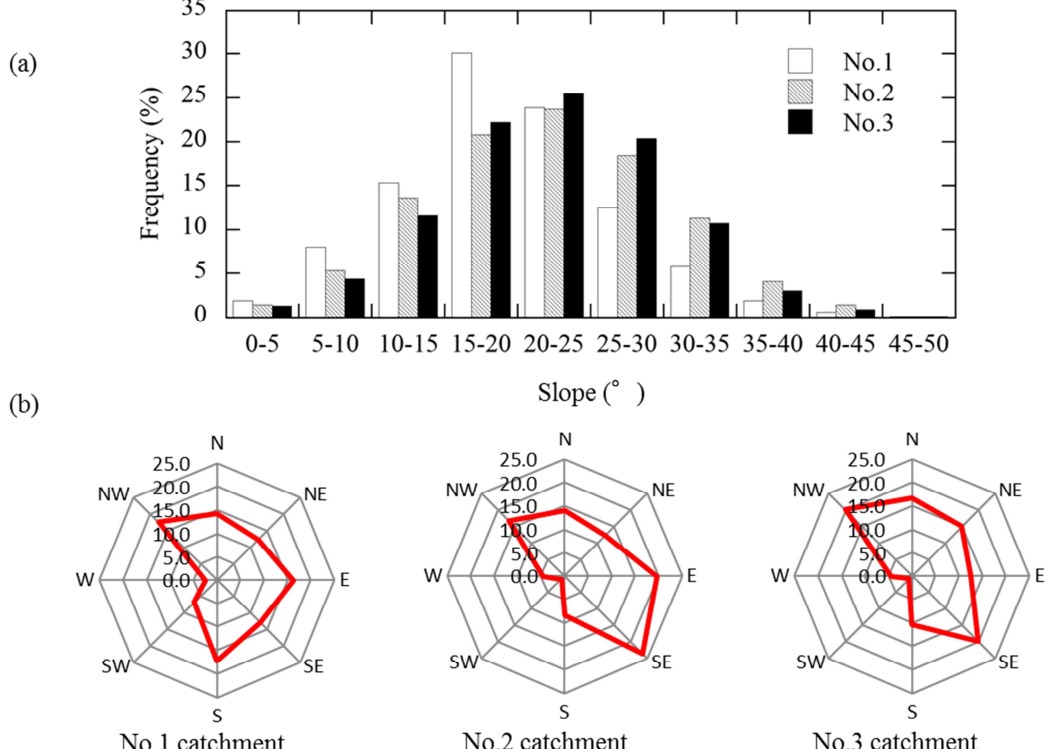

**Figure 2: Topographic characteristics in the Nagasaka experimental watershed. (a) Frequency of slope gradient and (b) frequency of slope direction.**





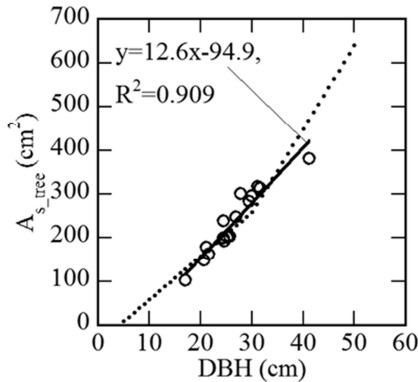

**Figure 3: Relationship between diameter at breast height (DBH) and sapwood area of each individual tree (As_tree). Dotted line is the relationship proposed by Tsurita et al. (2010).**





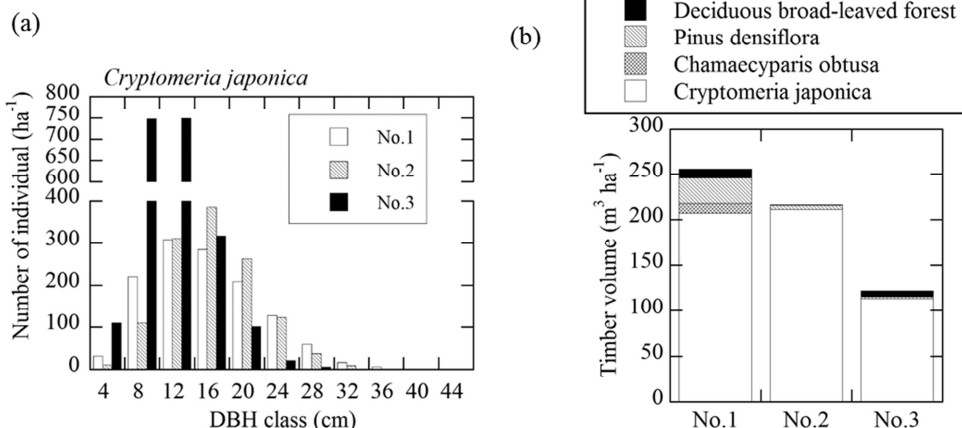

**Figure 4: Vegetation characteristics in the Nagasaka experimental watershed. (a) Frequency of diameter at breast height (DBH) class for *Cryptomeria japonica* stands and (b) ratios of timber volumes in catchment nos. 1, 2, and 3.**




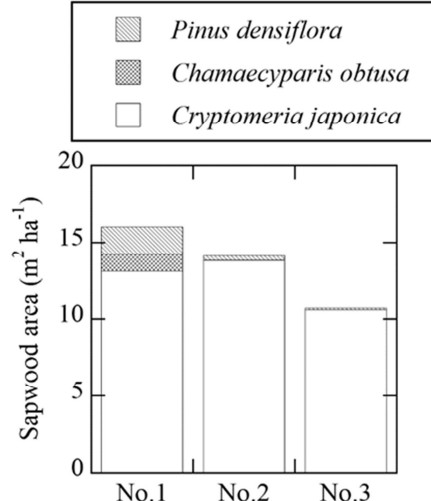

**Figure 5: Coniferous sapwood area (*Cryptomeria japonica*, *Chamaecyparis obtuse*, and *Pinus densiflora*) in catchment nos. 1, 2, and 3.**



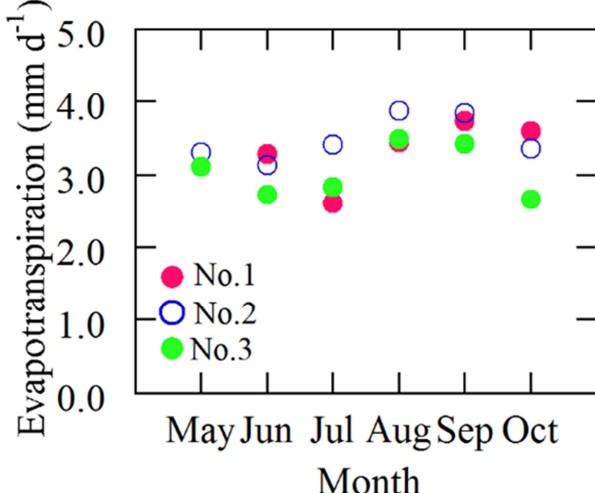

**Figure 6: Evapotranspiration in catchment nos. 1, 2, and 3.**





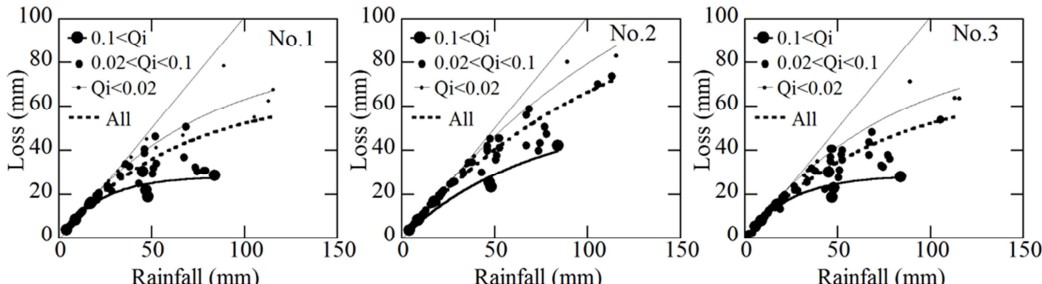

**Figure 7: Relationships between rainfall and water loss in catchment nos. 1, 2, and 3. The thin, thick, and dotted lines are the fitted relationships between rainfall and water loss when initial runoff ($Qi$) was $\leq 0.02$ mm h$^{-1}$, $\geq 0.1$ mm h$^{-1}$ and for all data.**

