# Peer review of "Effects of vegetation and soil on evapotranspiration, flow regime, and basin storage in three nearby catchments in northeast Japan"

_Hydrology and Earth System Sciences, 2016_

## Referee Comment (RC1) · Anonymous Referee #1 · 23 Dec 2016

The study carried out by Noguchi et al. consists of observations in three catchments of the Nagasake experimental watersheds in Japan. Diameter-at-breast height, sapwood area and soil properties were obtained. Evaporation was determined with a water-balance based method and basin storage was determined with an analytical relation between rainfall, stormflow and storage. The article is rather brief, and the added value and efforts of field campaigns may be clear. Nevertheless, I'd like to raise several issues that require some attention of the authors.

[Figure]

**1 General remarks**

In general, additional explanations are in some cases required. And even though the measurements and experiments are very valuable on its own, the authors merely don't fully manage to address the added value of the findings. To a large extend, the article is very descriptive, mentioning the numbers and outcomes of the measurements. The implications of these results don't become directly clear to the reader. The missing *Conclusions* section probably reflects this issue the most.

Some confusion is introduced in section *2.5.1 Evapotranspiration*. At first, it is described that total evaporation is estimated based on the assumption of no storage change between periods where q(t) and dq/dt are equal. This is a rather strong assumption, which may require some more supportive explanation. My first feeling is that I can draw many different hydrographs between t1 and t2 with the same slope and same discharge value, which may or may not cause a storage change, depending also on the rainfall in this period and evaporative energy. So please explain the method more detailed and add where these assumptions are based on, or add some references (not only written in Japanese) that do so.

Secondly, in lines 10-13, p5, a rather different approach is mentioned, not in line with the theory explained in the paragraph above. Instead of selecting start and end dates with the same values for q(t) and dq/dt, it is described here that start and enddates are selected based on rainfall rate and similar values of q(t). What is the reasoning behind this? In addition, why are periods < 10 days and >60 days excluded? Please add some more explanation in this paragraph why you made some choices.

After reading the discussion in 4.1, I would like to point at some aspects of the presented findings that are not discussed at all, but seem very interesting to me. It is pointed out in p8, lines 14-18, that the total evaporation is lower in catchment 3 due to lower sapwood area . Even though this is true, the total evaporation is just slightly smaller, whereas sapwood area is much smaller. It also mentioned that catchment 3
has a higher tree density. In addition, when looking at fig 4a, it seems that catchment 3 has more small trees, and catchments 1 and 2 more big trees. Therefore, my first guess would be that catchment 3 is a younger system, more effectively transpiring compared to catchments 1 and 2. Do you think this can play a role here? At least, it might be interesting to reflect on this.

I also wonder what is meant when 'basin storage' is discussed, in sections 3.3 and 4.2. At first, I thought you are discussing the maximum storage capacity of the basin, but eventually three different values are found based on the initial runoff. Therefore, is it the current active basin storage that is discussed? Please clarify this in your methods as well. In addition, how comparable are soil storage (which is a capacity) and, if meant so, the actual basin storage $S_B$?

Finally, throughout the paper, the terms transpiration and evapotranspiration are used. Please be aware that in some cases 'evapotranspiration' is used, whereas actually 'transpiration' is meant. For clarity, it might be better to use the term 'total evaporation' when the sum of interception evaporation, soil evaporation and transpiration is meant. I would like to point at Savenije (2004) for some more additional arguments to not use the term 'evapotranspiration'.

Concluding, the results presented in the paper are probably interesting for HESS, but the authors should put quite some effort in clarifying their methods and assumptions, and emphasize more on their key findings as well.

**2   Detailed comments**

Page 1, line29 –Page 2, line 2: you only refer to Japanese cases, for a more total picture, it might be good to refer to some other experimental watersheds as well (Hornbeck et al., 1997; Patric and Reinhart, 1971; Rothacher, 1970).

Page 2, line 16-17: "thus, percentage . . .water yield", this sentence seems a bit odd to me, rephrase?

Page 3, line 26: remove "of"

Page 5, line 30: which upper limit and towards what does the curve converge?

Page 5, line 34: please define if SB is the maximum basin storage (the capacity) or the current amount of storage in the system.

Page 5, line 25: It seems a result to me that there is an inflection at 12 and 72 hours after a storm. Is this the case for all three catchments? Maybe add a graph here as well.

Page 6, line 7: What do the percentages between brackets mean?

Page 6, line 19: You probably mean that the "mean transpiration is estimated by"

Page 7, lines 9-13: How is this basin storage determined? Do you fit Eq.10 to your data to obtain SB ? Please mention this in your methodology.

Page 7, line 15: 'that geology have' –> 'that geology has'

**3  references**

Hornbeck, J. W., Martin, C. W., and Eagar, C.: Summary of water yield experiments at Hubbard Brook Experimental Forest, New Hampshire, Canadian Journal of Forest Research, 27, 2043-2052, 10.1139/x97-173, 1997.

Patric, J. H., and Reinhart, K. G.: Hydrologic Effects of Deforesting Two Mountain Watersheds in West Virginia, Water Resources Research, 7, 1182-1188, 10.1029/WR007i005p01182, 1971.

Rothacher, J.: Increases in Water Yield Following Clear-Cut Logging in the Pacific Northwest, Water Resources Research, 6, 653-658, 10.1029/WR006i002p00653, 1970.

Savenije, H. H. G.: The importance of interception and why we should delete the term evapotranspiration from our vocabulary, Hydrological Processes, 18, 1507-1511, 10.1002/hyp.5563, 2004.

---

## Referee Comment (RC2) · T. de Boer-Euser (Referee) · 20 Jan 2017

The manuscript contains the description of a variety of measurements regarding vegetation (number of tree stands, tree diameter, sap wood area), soil properties (soil depth, pore sizes and hydraulic conductivity) and hydrological fluxes (precipitation and discharge). Based on these measurements the authors aim to link soil and vegetation characteristics to the hydrological behaviour of the catchment. I think that these kinds of measurements are important to increase our understanding of the hydrological functioning of catchments and that analyses based on measurement campaigns should be published. However, the results should be presented in a consistent way and relations between variables should be analysed carefully.

At the moment the manuscript is rather a presentation of measured data instead of an analysis of measurement results. In addition, the descriptions for different variables are not used in a consistent way. Below I have indicated some aspects which might help the authors to deepen their analysis and increase the value of the manuscript.

**General comments:**

**Relation between variables** The different measured categories of variables (i.e. vegetation, soil properties and hydrological behaviour) are presented separately. As the purpose of the manuscript is not only to present these variables, but to investigate the influence of vegetation and soil on the hydrological behaviour, it would be helpful to add some graphs in which these variables are plotted together.

**Presentation of results** The authors have presented a large amount of valuable data. However, I think that a different form of presentation can make the data more clear and interesting. For example, some paragraphs contain a lot of numbers, which might be more suitable to present in a table (e.g. P3L1-5) or even on a map (e.g. P6L6-14). In addition, I think that Table 2 can be replaced by a plot of a flow duration curve, maybe together with a plot of the hydrograph and measured preciptiation; this probably gives a clearer overview of the flow regime of the catchment. Further, it would be helpful for the reader to give a clear overview (maybe in a table) of the data collected for this study and the data used from other measurement campaigns, both with the period used.

**Effect on evapotranspiration** One of the aims of the manuscript is to investigate the influence on evapotranspiration. However, the calculation of evapotranspiration is based on strong assumptions and only presented as six monthly values for the entire measurement period. I think that these data are not sufficient to investigate the influence on evapotranspiration. What could improve the comparison is to actually present the transpiration that was calculated (P3L29), to present

more details about the ratios mentioned in P8L5-6 and to present an estimate for potential evaporation.

**Effect on basin storage** Another aim of the manuscript is to investigate the effect on basin storage. However, it is very unclear how basin storage is defined. The term would suggest it refers to the total storage capacity in the catchment; however, the soil water storage is often a bit higher, so the basin storage probably represents something else. Further, it is unclear whether basin storage is the actual amount of stored water (at which location?), a storage capacity, or even a flux (maybe ground water recharge?). On P5L1 it seems that again a different definition of storage is used. I think that the amount of (active) storage (capacity) in a catchment is very relevant, but different terms should be used in a consistent way.

**Conclusions** A conclusion section is missing, this again makes the manuscript more a description of data than an analysis of the influence of the measured variables on each other.

**Specific comments:**

The abstract contains too much detailed information and misses a clear conclusion regarding the influence of vegetation and soil on evaporation, discharge and storage as promised in the title.

The terms used to classify runoff (plentiful, ordinary, low, scanty) are not very common terms and might seem a bit vague and arbitrary to the reader. Therefore, they are especially not very suitable to use in the abstract.

Hydrology describes a cycle, this implies that there are no yields nor losses. Especially the term 'loss' is very confusing as the authors seem to use it for both interception evaporation as for groundwater recharge.

P2L3, this seems a very quick conclusion based on the previous sentence.

P3L2, which ratio is referred to here, that of slope to which other variable?

Why is the period for the analysis in section 2.1 different from that in 2.4?

2.3, choose one of the two: *soil storage* or *soil water storage*.

2.3, do the soil depth measurements represent the distance between the soil surface and the bedrock or to another impermeable layer?

2.3, I would use the same units (i.e. or *mm*, or *cm*) in the entire manuscript and especially within a table (Table 1), this also prevents the need for strange conversion factors as used in Eq.6.

P5L10, the second criteria seems to overrule the first.

2.5.3, this seems a standard method to separate base flow from storm flow, if this is the case, name it like that (with reference), otherwise include a figure explaining the procedure.

P5L29, is evaporation during the runoff event neglected?

P6L18, was the calculated transpiration smaller for catchment #3 as well?

P7L10, how does the basin storage follow from Figure 7?

P7L10, so the basin storage of #1 was smaller than that of #3 and both were smaller than for #2? If this is the case, it could be made clearer in the text.

P7L20, the previous paragraph seems to present that the basin storage in catchment #2 is a larger than for #1 and #3.
P8L5-6, how are these ratios determined? Do they originate from Iwaya et al. (2013)? If so, can they be assumed to be constant in time?

P9L3, this sentence seems to suggest that initial runoff and soil moisture content are basically the same, is this an appropriate assumption? I can imagine that initial runoff is determined by more factors than only soil moisture content.

P9L7, it would be interesting to elaborate this statement a bit further.

F1, some more indications of elevation would be helpful.

F5, consider combining this plot with Figure 4.

F6, for which year did you make this calculation?

F7, the dots for the individual events are difficult to distinguish, maybe try using different colours.

F7, there are four categories in the legend, but only three regression lines are presented. Why are not all categories presented with a regression line?

For investigating the influence of soil and vegetation on storage capacity and hydrological behaviour, these references might be of interest as well:

- Nijzink, R., Hutton, C., Pechlivanidis, I., Capell, R., Arheimer, B., Freer, J., Han, D., Wagener, T., McGuire, K., Savenije, H., and Hrachowitz, M.: The evolution of root-zone moisture capacities after deforestation: a step towards hydrological predictions under change?, Hydrol. Earth Syst. Sci., 20, 4775-4799, doi:10.5194/hess-20-4775-2016, 2016.
- de Boer-Euser, T., H. K. McMillan, M. Hrachowitz, H. C. Winsemius, and H. H. G. Savenije: Influence of soil and climate on root zone storage capacity, Water Resour. Res., 52, 2009–2024, doi:10.1002/2015WR018115, 2016.

**Technical comments:**

- Be consistent in using figures or words for indicating numbers, especially in P4L12 and P14L4;

- use consistent names or indications for the catchments, e.g. #1, #2, #3; instead of alternating 'catchment no. 1' and 'no. 1 catchment';

- P2L11, this sentence seems a bit strange;

- P2L16, *in* more detail;

- P2L25, consider using *catchment properties* or *catchment characteristics* instead of *hydrological factors*.

---

## Author Comment (AC1) · 24 Feb 2017

Dear Editor:

We greatly appreciate the efforts of the two reviewers and we have carefully considered and responded to all of their comments. Our responses are shown here (in blue typeset), including we made our changes and additions. Thank you all for your diligence. We would like to revise our manuscript based on the comments from reviewers. We hope that we have a chance to revise our manuscript.

Dear Anonymous Referee #1,

Thank you very much for your constructive comments concerning our manuscript entitled "Effects of vegetation and soil on evapotranspiration, flow regime, and basin storage in three nearby catchments in northeast Japan." Those comments are valuable and very helpful in revising and improving our paper. Our responses are shown here (in blue typeset).

1    General remarks

1)    In general, additional explanations are in some cases required. And even though the measurements and experiments are very valuable on its own, the authors merely don't fully manage to address the added value of the findings. To a large extend, the article is very descriptive, mentioning the numbers and outcomes of the measurements. The implications of these results don't become directly clear to the reader. The missing Conclusions section probably reflects this issue the most.

Response: We would like to add a Conclusions section in the revised manuscript.

2)    Some confusion is introduced in section 2.5.1 Evapotranspiration. At first, it is describe that total evaporation is estimated based on the assumption of no storage change between periods where q(t) and dq/dt are equal. This is a rather strong assumption, which may require some more supportive explanation. My first feeling is that I can draw many different hydrographs between t1 and t2 with the same slope and same discharge value, which may or may not cause a storage change, depending also on the rainfall in this period and evaporative energy. So please explain the method more detailed and add where these assumptions are based on, or add some references (not only written in Japanese) that do so. Secondly, in lines 10-13, p5, a rather different approach is mentioned, not in line with the theory explained in the paragraph above. Instead of selecting start and end dates with the same values for q(t) and dq/dt, it is described here that start and end dates are selected based on rainfall rate and similar values of q(t). What is the reasoning behind this? In addition, why are periods < 10 days and >60 days excluded? Please add some more explanation in this paragraph why you made some choices.

Response: We have revised the text and added some references in English (e.g., Linsley et al., 1982; Kosugi and Katsuyama, 2007) to clarify the short-time period water-budget method as follows:

Mean basin evapotranspiration was estimated using the short-time period water-budget method (SPWB) reported by Linsley et al. (1982). The relationship between water storage in basin $S(t)$ and discharge rate $q(t)$ can be written using Eq. (7):

$$S(t) = f[q(t), dq/dt] \tag{7}$$

Assuming the water storage values $S(t_1)$ and $S(t_2)$ are equal when $q(t)$ and $dq/dt$ become equal at times $t_1$ and $t_2$, the change in water storage $dS/dt = S(t_2) - S(t_1)$ is zero. Then, $ET$ is calculated using Eq. (8):

$$ET = P - Q = \int_{t_2}^{t_1} p(t)dt - \int_{t_2}^{t_1} q(t)dt \tag{8}$$

Where $P$ and $Q$ are total precipitation and total discharge from $t_1$ to $t_2$, and $p(t)$ and $q(t)$ are the rainfall intensity and discharge rate, respectively. This method was used in estimating seasonal variations of the $ET$ by Hamon (1961). The following procedure was used to determine the water budget periods for $t_1$ and $t_2$ (Suzuki, 1985; Kosugi and Katsuyama, 2007).

1) Find all days such that no rain fell during the previous two days as potential beginning or ending days for the hydrological period.

2) Select all pairs from these candidate days such that the daily runoff rates were within $a$% of the average runoff rate for both days.

3) Omit pairs of days where the period covered was less than $b$ days or more than $c$ days.

The first condition is intended to remove rapid flow, which has large $dq/dt$. The second condition is to find the equivalent $q$ at times $t_1$ and $t_2$. In the third condition, evapotranspiration varies largely when the water balance period is too short. In addition, it is not suitable to evaluate seasonal variation in evapotranspiration if the water budget period is long. On the other hand, Noguchi et al. (2004) considered a canopy water storage capacity based on the interception observation in a tropical rain forest to estimate $ET$ in the first condition. This is because they obtain an adequate number of samples for the water budget periods for $t_1$ and $t_2$. The canopy water storage capacity value for *C. japonica* was 2.22 mm (Saito et al., 2013). Therefore, rainfall $\leq$ 2.0 mm d$^{-1}$ has no opportunity to become soil moisture. Then, instead of specifying the first condition as indicated above, we adopt the following condition in this study: Find all days such that $\leq$ 2.0 mm d$^{-1}$ rain fell during the previous two days as potential beginning or ending days for the hydrological period. In addition, those conditions ($a$, $b$, and $c$) in previous studies were set by trial and error to obtain large samples. We also set those conditions in this study by trial and error for calculation of $ET$ (Table 1). If more than one value was obtained on the same day, estimates were averaged. The $ET$ values obtained by SPWB were

averaged monthly for three years.

Table 1 Criteria for water budget period selection

| Reference | $a$ (%) | $b$ (days) | $c$ (days) |
|---|---|---|---|
| Kosugi and Katsuyama (2007) | 2 | 8 | 60 |
| Murakami et al. (2000) | 5 | 8 | 60 |
| Noguchi et al. (2004) | 5 | 8 | 120 |
| Suzuki (1985) | 2 | 8 | 60 |
| Zulkifli et al. (2008) | 5 | 8 | 120 |
| This study | 5 | 10 | 60 |

3) After reading the discussion in 4.1, I would like to point at some aspects of the presented findings that are not discussed at all, but seem very interesting to me. It is pointed out in p8, lines 14-18, that the total evaporation is lower in catchment 3 due to lower sapwood area. Even though this is true, the total evaporation is just slightly smaller, whereas sapwood area is much smaller. It also mentioned that catchment 3 has a higher tree density. In addition, when looking at fig 4a, it seems that catchment 3 has more small trees, and catchments 1 and 2 more big trees. Therefore, my first guess would be that catchment 3 is a younger system, more effectively transpiring compared to catchments 1 and 2. Do you think this can play a role here? At least, it might be interesting to reflect on this.

Response: The mean ratios of throughfall to rainfall in each NEW catchment are 82.1% (no.1), 80.1% (no.2), and 80.3% (no.3), respectively (Iwaya et al. 2013). In general, the ratio of stemflow to rainfall is smaller than the ratio of throughfall to rainfall. These results suggest that the difference in interception loss among the three catchments is relatively small. Transpiration is not measured directly, but we believe that there is a relatively smaller amount of transpiration occurring in the catchment 3 than in the other two catchments. Because the sapwood area (10.7 $m^2$ $ha^{-1}$) in catchment no. 3 were smaller than those in the other two catchments (no. 1: 255.7 $m^3$ $ha^{-1}$; 16.0 $m^2$ $ha^{-1}$, no. 2: 216.5 $m^3$ $ha^{-1}$; 14.2 $m^2$ $ha^{-1}$). We estimated the result based on catchment scale. In addition, tree census has been done in 2002 and 2005 at plot scale (0.2ha). The change in stand volume for three years in catchment no.3 (41.8 $m^3$ $ha^{-1}$) was also smaller than those in the other two catchments (no.1: 80.5 $m^3$ $ha^{-1}$, no.2: 89.3 $m^3$ $ha^{-1}$). This result more strongly supports that amount of transpiration in catchment no.3 is small. We would like to add data supporting this belief in revised manuscript.

4) I also wonder what is meant when 'basin storage' is discussed, in sections 3.3 and 4.2. At first, I thought you are discussing the maximum storage capacity of the basin, but eventually three different values are found based on the initial runoff. Therefore, is it the current active basin

storage that is discussed? Please clarify this in your methods as well. In addition, how comparable are soil storage (which is a capacity) and, if meant so, the actual basin storage SB?

Response: The basin storage ($S_B$) was calculated as the relationship between basin water loss and rainfall using the approximate equation (10). The basin storage depended on the initial runoff. We use the term "retention capacity", not the term "water loss", and the term "total retention capacity" instead of "basin storage,$S_B$." We then revised the sentences as follows:

Retention capacity, $C_R$ (mm), was computed based on stormflow, $SF$ (mm), and precipitation, $P$ (mm), for every storm event using Eq. (9):

$$C_R = P\text{-}SF \tag{9}$$

The relationship between $C_R$ and $P$ shows that $R_C$ increases with increase in $P$, and approximates the envelope curve. The curve was defined as a retention curve using Eq. (10):

$$L = TC_R(1 - e^{-kP}) \tag{10}$$

Where $TC_R$ is the total retention capacity of the catchment (mm) and $k$ is a constant (Endo 1985; Fujieda 2007).

$TC_R$ depended on the initial runoff. We redrew the figure to understand the results.

Then, we compared the size of $TC_R$ and soil storage among catchment nos. 1, 2, and 3.

Soil water storage is calculated based on the laboratory experiments for the soil physical properties and on results of soil depth measurements in the catchments. Soil water storage is constant. The total retention capacity is calculated based on the observations of rainfall and discharge and depends on the soil moisture conditions; it is thus different from the soil water storage.

5) Finally, throughout the paper, the terms transpiration and evapotranspiration are used. Please be aware that in some cases 'evapotranspiration' is used, whereas actually 'transpiration' is meant. For clarity, it might be better to use the term 'total evaporation' when the sum of interception evaporation, soil evaporation and transpiration is meant. I would like to point at Savenije (2004) for some more additional arguments to not use the term 'evapotranspiration'.

Response: Thank you for your comment. Linsley et al. (1982) described water-budget determination of mean basin evapotranspiration. We calculated ET in this paper using this method. We normally use the term "evapotranspiration" instead of the term "total evaporation" (e.g., Murakami et al, 2001; Kosugi and Katsuyama, 2006). In forest hydrology, the loss of water is commonly termed evapotranspiration when it refers to the total loss of water in the vapor state (Chang, M, 2006). Then, in the revised paper, we referenced these articles and described ET as follows: Evapotranspiration is the sum of several components (Hewlett, 1986):

$$ET = T + It + Es + Eo$$

where T is transpiration, It is canopy interception, Es is evaporation from soil, and Eo is open-water evaporation.

2    Detailed comments

1)   Page 1, line29 –Page 2, line 2: you only refer to Japanese cases, for a more total picture, it might be good to refer to some other experimental watersheds as well (Hornbeck et al., 1997; Patric and Reinhart, 1971; Rothacher, 1970).
     Response: Thank you for your suggestion. We would like to revise the Introduction section to include the above papers.

2)   Page 2, line 16-17: "thus, percentage . . .water yield", this sentence seems a bit odd to me, rephrase?
     Response: We would like to revise the sentence as follows: Thus, we need more detailed information on not only percentage of forest cover but also location of forest cut and vegetation status to evaluate water yield.

3)   Page 3, line 26: remove "of"
     Response: Thank you for your comment. We will remove "of" in the revised manuscript as suggested.

4)   Page 5, line 30: which upper limit and towards what does the curve converge?
     Response: We fitted Eq.10 (modified exponential curve) to the data according to the initial runoff to obtain $S_B$. We have revised the Methods and Figure 7 to explain how we arrived at $S_B$.

[Figure]

Figure 7

5) Page 5, line 34: please define if SB is the maximum basin storage (the capacity) or the current amount of storage in the system.

Response: The basin storage ($S_B$) was calculated as the relationship between retention capacity (previously we used water loss instead of retention capacity) and rainfall using the approximate equation (10). $S_B$ depended on the initial runoff. Therefore, $S_B$ is the current amount of storage in the system.

6) Page 5, line 25: It seems a result to me that there is an inflection at 12 and 72 hours after a storm. Is this the case for all three catchments? Maybe add a graph here as well.

Response: We would like to add a figure, which explain how to calculate stormflow. The figure caption is "this diagram defines storm runoff as described by the storm hydrograph."

7) Page 6, line 7: What do the percentages between brackets mean?

Response: We would like to revise the sentence as follows: The ratios of evergreen coniferous tree stand volume in catchment nos. 1, 2, and 3 were 96.0%, 99.6%, and 94.7%, respectively. The ratios of *C. japonica* stand volume in catchment nos. 1, 2, and 3 were 80.9%, 97.5%, and 92.8%, respectively.

8) Page 6, line 19: You probably mean that the "mean transpiration is estimated by"

Response: Thank you for your comment. We would like to revise the sentence as follows: The amount of mean transpiration from a catchment is the product of mean sap flux density and total sapwood area.

9) Page 7, lines 9-13: How is this basin storage determined? Do you fit Eq.10 to your data to obtain SB ? Please mention this in your methodology.

Response: Yes, we fitted Eq.10 to the data to obtain $S_B$. We have revised Figure 7 (please see comment #4) to explain how to obtain $S_B$ using the relationship between retention capacity (previously we used water loss instead of retention capacity) and rainfall. We use the total retention capacity ($TC_R$) in a catchment instead of basin storage.

10) Page 7, line 15: 'that geology have' –> 'that geology has'

Response: The necessary changes will be incorporated in the revised manuscript.

Additional references:

Kosugi, Y., Katsuyama, M.: Evapotranspiration over a Japanese cypress forest. II Comparison of the

eddy covariance and water budget methods, J. Hydrology, 334, 305–311, doi.org/10.1016/j.jhydrol.2006.05.025

Saito, T., Matsuda, H., Komatsu, M., Xiang, Y., Takahashi, A., Shinohara, Y., Otsuki, K.: Forest canopy interception loss exceeds wet canopy evaporation in Japanese cypress (Hinoki) and Japanese cedar (Sugi) plantations, Journal of Hydrology, 507, 287–299, doi:680 10.1016/j.jhydrol.2013.09. 053

Linsley, P.K., Kohler, M.A. and Paulhus, J.L.: Hydrology for Engineers 3rd ed., 508pp. McGraw-Hill, New York, 1982.

Chang, M.: Forest Hydrology. an Introduction to water and forests. 2nd ed., 474pp. Taylor & Francis, Boca, Raton, London, New York, 2006

Hamon, W. R., 1961. Estimating Potential Evapotranspiration. ASCE. J. Hydraulics Division, 87:107-120.

Zulkifli Y., Chong, M.H., Geoffery J. G., Ayob K.: Estimation of evapotraspiration in oil palm catchments by short-time period water-budget method. Malaysian Journal of Civil Engineering, 20(1), 160-174, 2008.

Dear Dr. T. de Boer-Euser,

Thank you very much for your constructive comments concerning our manuscript entitled "Effects of vegetation and soil on evapotranspiration, flow regime, and basin storage in three nearby catchments in northeast Japan." Those comments are all valuable and very helpful in revising and improving our paper; they also provide important guidance to us in our research. Our responses are shown here (in blue typeset).

The manuscript contains the description of a variety of measurements regarding vegetation (number of tree stands, tree diameter, sap wood area), soil properties (soil depth, pore sizes and hydraulic conductivity) and hydrological fluxes (precipitation and discharge). Based on these measurements the authors aim to link soil and vegetation characteristics to the hydrological behaviour of the catchment. I think that these kinds of measurements are important to increase our understanding of the hydrological functioning of catchments and that analyses based on measurement campaigns should be published. However, the results should be presented in a consistent way and relations between variables should be analysed carefully.

At the moment the manuscript is rather a presentation of measured data instead of an analysis of measurement results. In addition, the descriptions for different variables are not used in a consistent way. Below I have indicated some aspects which might help the authors to deepen their analysis and increase the value of the manuscript.

General comments:

**Relation between variables**: The different measured categories of variables (i.e. vegetation, soil properties and hydrological behaviour) are presented separately. As the purpose of the manuscript is not only to present these variables, but to investigate the influence of vegetation and soil on the hydrological behaviour, it would be helpful to add some graphs in which these variables are plotted together.

Response: Thank you for your comment. We would like to add a figure in revised manuscript.

[Figure]

Figure

**Presentation of results**: The authors have presented a large amount of valuable data. However, I think that a different form of presentation can make the data more clear and interesting. For example, some paragraphs contain a lot of numbers, which might be more suitable to present in a table (e.g. P3L1-5) or even on a map (e.g. P6L6-14). In addition, I think that Table 2 can be replaced by a plot of a flow duration curve, maybe together with a plot of the hydrograph and measured preciptiation; this probably gives a clearer overview of the flow regime of the catchment. Further, it would be helpful for the reader to give a clear overview (maybe in a table) of the data collected for this study and the data used from other measurement campaigns, both with the period used.

Response: Although we consider that that Table 2 shows the differences in the flow regime between the three catchments, we do agree with your suggestion. Therefore, in addition to Table 2, we would add also the figure showing hyetographs, hydrographs and flow duration curves. Soil physical properties are shown in Table 1. We would like to add a table showing the results of tree census in revised manuscript.

[Figure]

Figure    Hyetographs, hydrographs and flow duration curves

**Effect on evapotranspiration**: One of the aims of the manuscript is to investigate the influence on evapotranspiration. However, the calculation of evapotranspiration is based on strong assumptions and only presented as six monthly values for the entire measurement period. I think that these data are not sufficient to investigate the influence on evapotranspiration. What could improve the comparison is to actually present the transpiration that was calculated (P3L29), to present more details about the ratios mentioned in P8L5-6 and to present an estimate for potential evaporation.

Response: Evapotranspiration is estimated using the short-time period water-budget (SPWB) method. We would like to add references to associated papers and revise our sentences to clarify use of this method. The Nagasaka experimental watershed is located in a snow cover area and it is thus not suitable to use the SPWB method to estimate evapotranspiration during the snow season. Furthermore, amount of transpiration are also neglected during the snow season. This is the reason why we have presented as six monthly values for the no snow season.

Tree census has been done in 2002 and 2005 at plot scale (0.2 ha). The change in stand volume for three years in catchment no.3 (41.8 $m^3$ $ha^{-1}$) was also smaller than those in the other two catchments (no.1: 80.5 $m^3$ $ha^{-1}$, no.2: 89.3 $m^3$ $ha^{-1}$). This result more strongly supports that amount of transpiration in catchment no.3 is small. We would like to add data supporting this belief in revised manuscript.

Throughfall was observed for 167 days in 2004, for 197 days in 2005, and for 206 days in 2006: the ratios of throughfall to rainfall were between 80.8 and 83.4% (mean: 82.0%) in #1, 78.7 and 80.9% (80.1%) in #2, and 77.8 and 81.9% (mean: 80.2%) in #3 (Iwaya et al., 2013).

**Effect on basin storage**: Another aim of the manuscript is to investigate the effect on basin storage. However, it is very unclear how basin storage is defined. The term would suggest it refers to the total storage capacity in the catchment; however, the soil water storage is often a bit higher, so the basin storage probably represents something else. Further, it is unclear whether basin storage is the actual amount of stored water (at which location?), a storage capacity, or even a flux (maybe ground water recharge?). On P5L1 it seems that again a different definition of storage is used. I think that the amount of (active) storage (capacity) in a catchment is very relevant, but different terms should be used in a consistent way.

Response: Thank you for your comment. We use the term "retention capacity" instead of "water loss", and also use the term "total retention capacity" instead of "basin storage." Soil water storage is calculated based on the laboratory experiments for the soil physical properties and on results of soil depth measurements in the catchments. Soil water storage is constant. The total retention capacity is calculated based on the observations of rainfall and discharge and depends on the soil moisture conditions; it is thus different from the soil water storage.

**Conclusions**: A conclusion section is missing, this again makes the manuscript more a description of data than an analysis of the influence of the measured variables on each other.

Response: We would like to add a Conclusions section in the revised manuscript.

Specific comments:

The abstract contains too much detailed information and misses a clear conclusion regarding the influence of vegetation and soil on evaporation, discharge and storage as promised in the title.

Response: We would like to revise the abstract so that it delivers a clear conclusion and have also deleted superfluous information.

The terms used to classify runoff (plentiful, ordinary, low, scanty) are not very common terms and might seem a bit vague and arbitrary to the reader. Therefore, they are especially not very suitable to use in the abstract.

Response: We now delete the terms "plentiful, low, and scanty runoffs" from the abstract.

Hydrology describes a cycle, this implies that there are no yields nor losses. Especially the term 'loss' is very confusing as the authors seem to use it for both interception evaporation as for groundwater recharge.

Response: We would like to use the term "retention capacity" instead of "water loss."

P2L3, this seems a very quick conclusion based on the previous sentence.

Response: We would like to revise this sentence as follows, "These studies implied that vegetation is one of the important factors determining evapotranspiration and flow regime."

P3L2, which ratio is referred to here, that of slope to which other variable?

Response: We show the ratio of slope direction.

Why is the period for the analysis in section 2.1 different from that in 2.4?

Response: Because air temperature was not observed prior to 2006.

2.3, choose one of the two: soil storage or soil water storage.

Response: Thank you for your comment. We have used soil water storage in the paper.

2.3, do the soil depth measurements represent the distance between the soil surface and the bedrock or to another impermeable layer?

Response: Soil depth was measured using a Hasegawa soil sampler (Daitou Techno Green, Inc., Tokyo, Japan). The number of measuring points was 57, 56, and 57 in catchments nos. 1, 2, and 3, respectively (total of 170 points). Soil depth could be measured up to 1m using the sampler. Table 1 describes the results at 2, 15, and 5 sampling points at soil depths > 100 cm soil deep in catchment nos. 1, 2, and 3, respectively. These points mean until B layer. Other points mean until C layer.

2.3, I would use the same units (i.e. or *mm*, or *cm*) in the entire manuscript and especially within a table (Table 1), this also prevents the need for strange conversion factors as used in Eq.6.

Response: Thank you for your comment; we have now used the same units throughout the entire manuscript.

P5L10, the second criteria seems to overrule the first.

Response: Noguchi et al. (2004) considered a canopy water storage capacity based on the interception observation in a tropical rain forest to estimate ET in the first condition. This is because they obtain an adequate number of samples for the water budget periods for t1 and t2. The canopy water storage capacity value for C. japonica was 2.22 mm (Saito et al., 2013). Therefore, rainfall $\leq 2.0$ mm d$^{-1}$ has no opportunity to become soil moisture. Then, instead of specifying the first condition as indicated above, we adopt the following condition in this study: Find all days such that $\leq 2.0$ mm d$^{-1}$ rain fell during the previous two days as potential beginning or ending days for the hydrological period. We have revised the sentences in better

detail (please see Response 1 for Anonymous Referee #1).

2.5.3, this seems a standard method to separate base flow from storm flow, if this is the case, name it like that (with reference), otherwise include a figure explaining the procedure.

Response: We would like to add a figure to explain the procedure.

P5L29, is evaporation during the runoff event neglected?

Response: We are describing "water loss, L (mm)" in this section, not evaporation. We would like to use "retention capacity" instead of "water loss."

P6L18, was the calculated transpiration smaller for catchment #3 as well?

Response: We have not measured transpiration directly. However, we would like to add data relating to the change in stand volume, which supports the result in the revised manuscript.

P7L10, how does the basin storage follow from Figure 7?

Response: We have now re-drawn Figure 7 (please see response no.4 in detailed comments for Anonymous Referee #1) We fitted Eq.10 (modified exponential curve) to the data according to the initial runoff to obtain $S_B$.

P7L10, so the basin storage of #1 was smaller than that of #3 and both were smaller than for #2? If this is the case, it could be made clearer in the text. P7L20, the previous paragraph seems to present that the basin storage in catchment #2 is a larger than for #1 and #3.

Response: Yes, the basin storage in catchment #2 is the largest among the three catchments; this is because soil water storage in catchment #2 is also the largest among the three catchments.

P8L5-6, how are these ratios determined? Do they originate from Iwaya et al. (2013)? If so, can they be assumed to be constant in time?

Response: Throughfall was observed for 167 days in 2004, for 197 days in 2005 and for 206 days in 2006. As shown above, the ratios of throughfall to rainfall were from 80.8 to 83.4% (mean: 82.0%) in #1, from 78.7 to 80.9% (80.1%) in #2, and from 77.8 to 81.9% (mean: 80.2%) in #3 (Iwaya et al., 2013).

P9L3, this sentence seems to suggest that initial runoff and soil moisture content are basically the same, is this an appropriate assumption? I can imagine that initial runoff is determined by more factors than only soil moisture content.

Response: It depends on size of catchment. Initial runoff can be an index of soil moisture in small

catchments (e.g., dozens of hectares).

P9L7, it would be interesting to elaborate this statement a bit further.

Response: We added the following sentence: "This result suggest that the initial runoff could be a parameter to predict the total retention capacity in the current condition."

F1, some more indications of elevation would be helpful.

Response: We would like to add some more indications of elevation in Figure 1.

F5, consider combining this plot with Figure 4.

Response: We would like to combine Figure 4 and Figure 5 in the revised manuscript.

F6, for which year did you make this calculation?

Response: The ET values obtained by SPWB were averaged monthly for three years. We have explained this in the "Method" and would like to revise the figure to show the differences in the revised nanuscript.

F7, the dots for the individual events are difficult to distinguish, maybe try using different colours.

Response: We revised the figure to distinguish the differences.

F7, there are four categories in the legend, but only three regression lines are presented. Why are not all categories presented with a regression line?

Response: We now show four regression lines for each catchment in the revised figure. We have now re-drawn Figure 7 (please see response no.4 in detailed comments for Anonymous Referee #1).

For investigating the influence of soil and vegetation on storage capacity and hydrological behaviour, these references might be of interest as well:

· Nijzink, R., Hutton, C., Pechlivanidis, I., Capell, R., Arheimer, B., Freer, J., Han, D., Wagener, T., McGuire, K., Savenije, H., and Hrachowitz, M.: The evolution of root-zone moisture capacities after deforestation: a step towards hydrological predictions under change?, Hydrol. Earth Syst. Sci., 20, 4775- 4799, doi:10.5194/hess-20-4775-2016, 2016.

· de Boer-Euser, T., H. K. McMillan, M. Hrachowitz, H. C. Winsemius, and H. H. G. Savenije: Influence of soil and climate on root zone storage capacity, Water Resour. Res., 52, 2009–2024, doi:10.1002/2015WR018115, 2016.

Response: Thank you for introducing interesting references. I'd like to quote these in the revised manuscript.

Technical comments:

· Be consistent in using figures or words for indicating numbers, especially in P4L12 and P14L4;

Response: We have added information in Table 1 for clarification, as follows: † Classification of soil layer is based on the Hasegawa soil sampler. † † Classification of soil layer is based on the observation of profile at soil pit.

· use consistent names or indications for the catchments, e.g. #1, #2, #3; instead of alternating 'catchment no. 1' and 'no. 1 catchment';

Response: We have changed the names of catchment to #1, #2, and #3 instead of catchment nos. 1, 2, and 3 as names for the catchments.

· P2L11, this sentence seems a bit strange;

Response: The relationship between increases in water yield following reduction of forest cover and the ratio of reduction in cover was evaluated using the paired catchment studies (Bosch and Hewlett 1982; Brown et al. 2005).

· P2L16, in more detail;

Response: We revised the sentence as follow, "Thus, percentage changes in forest cover and vegetation status (*e.g.* stand volume, sapwood area) at a catchment scale are needed to evaluate water yield."

· P2L25, consider using catchment properties or catchment characteristics instead of hydrological factors.

Response: Thank you for your comment. We have now used "catchment properties" instead of "hydrological factors."